# Frailty and Diminished Human Relationships Are Associated with Poor Sleep Quality in Japanese Older Adults: A Cross-Sectional Study

**DOI:** 10.3390/geriatrics8050091

**Published:** 2023-09-11

**Authors:** Hitomi Matsuda, Thomas Mayers, Naoki Maki, Akihiro Araki, Sachie Eto

**Affiliations:** 1Graduate School of Comprehensive Human Sciences, Department of Human Care Science, University of Tsukuba, 1-1-1 Tennodai, Tsukuba 305-8575, Ibaraki, Japan; eto.sachie.gp@un.tsukuba.ac.jp; 2Department of Health Services Research, Institute of Medicine, University of Tsukuba, 1-1-1 Tennodai, Tsukuba 305-8575, Ibaraki, Japan; mayers@md.tsukuba.ac.jp; 3Medical English Communications Center, Institute of Medicine, University of Tsukuba, 1-1-1 Tennodai, Tsukuba 305-8575, Ibaraki, Japan; 4Faculty of Rehabilitation, R Professional University of Rehabilitation, 2-12-31 Kawaguchi, Tsuchiura 300-0032, Ibaraki, Japan; maki@a-ru.ac.jp; 5Community Health Nursing Department, Oita University of Nursing and Health Sciences, 2944-9 Megusuno, Oita-City 870-1201, Oita, Japan; araki@oita-nhs.ac.jp

**Keywords:** Japanese older adults, sleep quality, frailty, human relationships, conversation

## Abstract

The purpose of this study was to examine the association between sleep quality, frailty, and human relationships in Japanese older adults (aged 65 years and above, excluding those certified as requiring long-term care). This cross-sectional study used a questionnaire survey to gather demographic information, data on frequency of conversation and conversation partners, and employed the following validated instruments: Kihon Checklist (KCL), a Japanese instrument used to determine the care needs and frailty of older adults; the Dysphagia Risk Assessment for Community-Dwelling Elderly (DRACE) scale; Japanese versions of Pittsburgh Sleep Quality Index (PSQI-J); the Geriatric Depression Scale-15 (GDS-15-J); and the University of California Los Angeles Scale (UCLA-J), an instrument to assess loneliness in older adults. The 500 respondents were divided into two groups based on sleep quality (PSQI-J): low sleep quality group (*n* = 167, 33.4%) and high sleep quality group *(n* = 333, 66.6%). Our analyses showed that the low sleep quality group had a KCL score of 5.55 ± 2.47, which indicated frailty. Binomial logistic regression analysis identified age, number of diseases under treatment, DRACE, GDS-15-J, and conversation frequency and discussion partner for important matters as factors (*p* < 0.05) associated with poor sleep quality. These factors could help enhance the detection of frailty and predictability of caregiving needs.

## 1. Introduction

In recent times, Japan has witnessed significant shifts in its social landscape, particularly among its older population. Presently, over 60% of older adults in the country live alone or in older households with older couples. Reports have indicated that with this evolving societal context, human relationships have become increasingly weak, spanning from interactions with neighbors as far as parent-child bonds [1]. Diminished social relationships have been shown to be associated with poor sleep quality [2,3]. Sleep disorders are said to affect 30% of older adults in Japan, and poor sleep quality is associated with a decline in physical, mental, and social functioning, and a negative cycle of loneliness and isolation, depression, and suicide has been noted [4]. The effects of insomnia in older adults are far-reaching and the association between insomnia and frailty have increasingly been documented [5,6,7].

Diminished social relationships are becoming an even more pressing issue in the aftermath of the COVID-19 pandemic, where many older adults have faced months if not years of social distancing and social isolation [8]. This backdrop has led to an increased focus in a concept known as “social frailty”, which is used to refer to the situation of declining social function among older adults. Despite the recognition of “social frailty” as a phenomenon, a unified definition or set of components has been elusive. Fujiwara operationally defines social frailty as “a state of increased vulnerability to participation in social activities and interaction,” and lists “going out and interacting” as assessment items [9,10]. Makizako and colleagues expanded on this by offering the following evaluation items: (1) less frequent going out compared to the previous year, (2) not visiting friends’ houses, (3) not being useful to friends and family, (4) living alone, and (5) not talking to someone every day [11]. They posit that fulfilling two or more of these criteria indicates the presence of social frailty [11]. Notably, some of these criteria delve into topics (such as household composition and conversation) that are not covered by existing assessment tools. Given the prevailing circumstances among older adults, these additional items would be useful for comprehensive assessment of social functioning and, therefore, social frailty. 

The Kihon Checklist (KCL), developed by the Japanese Ministry of Health, Labor, and Welfare [12] serves as a reliable screening tool to identify high-risk individuals necessitating nursing care [13,14]. While the KCL’s effectiveness in assessing physical and mental frailty in older adults is widely acknowledged, [15] it lacks an assessment of sleep quality. In 2020, in an effort to integrate health services for older adults and care prevention, a health checkup questionnaire for older adults was developed with 15 items over 10 categories [16]. While some of the items on the questionnaire are common to the KCL, and their combined use is thought to improve the ability to discriminate frailty, the questionnaire also lacked any sleep-related items [16]; thus, issues regarding efficient assessment methods and provision systems for early detection of frailty among older adults remain [17].

Given the need to accurately assess the care needs of older adults and the intricate web of factors that can influence their health and wellbeing, the objective of the current study was to investigate the interplay between conversation and other elements that are associated with sleep quality. This study targeted independently living Japanese older adults with the intention of contributing to the prevention of long-term care dependency. Recognizing the evolving social dynamics and the multifaceted nature of frailty, this study sought to investigate potential connections between conversation habits and sleep quality among this demographic.

## 2. Materials and Methods

### 2.1. Study Participants

For this study, Rakuten Insight (Rakuten Insight Global, Inc., Tokyo, Japan), an established entity with a proven track record in Internet-based research and a registered monitor base of 2.2 million, was commissioned. The study cohort consisted of 500 independently living older adults aged 65 years or older who were registered as monitors with Rakuten Insight. The sample size was determined by first considering the elderly population using the Internet. In the fiscal year 2021, Internet usage among individuals aged 65 and above was approximately 53.4% [18]. Utilizing this percentage, along with a 95% confidence level and a 5% margin of error, the required population proportion was calculated to be 382 participants. Furthermore, the statistical analyses for this study involved *t*-tests with men and women aged 65 and older. Consequently, by assuming a moderate effect size (d = 0.5), an alpha of 0.05, a power of 0.95, and an allocation ratio of 1, each group necessitated 105 participants (210 in total) (using G*Power for Windows 3.1.9.7, RRID:SCR_013726, Dusseldorf, Germany). Consequently, a sample size of 500 was established, surpassing all criteria.

Participant informed consent was obtained prior to conducting the research. Participants were required to abide by the personal information protection regulations (https://insight.rakuten.co.jp/company/privacy.html; accessed on 21 June 2021) and meet the following inclusion conditions: (1) no “care-needs certification” requiring 1–5 years of nursing care, (2) absence of a dementia diagnosis, and (3) capacity to comprehend and act upon instructions and responses. To protect the participants’ personal information, Rakuten Insight provided anonymized participant data to the principal investigator and acted as an intermediary when participants wished to withdraw consent or had inquiries about the study. Details of the informed consent procedures are given in Appendix A.

The survey was conducted from 1 to 20 June 2021, in accordance with the guidelines of the Declaration of Helsinki and after being approved by the ethics committee of the Institute of Medicine, University of Tsukuba (Approval No. 1623-5).

### 2.2. Research Design and Survey Items

Considering the COVID-19 pandemic situation and government guidelines, a cross-sectional study utilizing an Internet-based questionnaire survey was conducted to mitigate the risk of infection transmission.

The survey included (1) demographic items (age, sex); (2) ongoing medical conditions (diseases) and their quantity; (3) frequency of daily conversation, and discussion partners for important matters; and (4) validated questionnaire instruments: the KCL, the Japanese versions of Pittsburgh Sleep Quality Index (PSQI-J); Dysphagia Risk Assessment for Community-Dwelling Elderly (DRACE); Geriatric Depression Scale-15 (GDS-15-J); and the Japanese version of University of California Los Angeles Scale (UCLA-J), an instrument to assess loneliness in older adults. The validity of the PSQI-J [19], DRACE [20], GDS-15-J [21], and UCLA-J [22], has been supported in previous validation studies by Cronbach’s alpha coefficients of 0.77, 0.88, and 0.84, respectively.

#### 2.2.1. Assessment of Sleep Quality

The PSQI-J has been used to evaluate and screen for sleep disorders in older adults [19]. This scale assesses sleep status over the previous month. Each item is scored from 0 to 3 points, resulting in a score range of 0–21 points. The threshold for identifying sleep disturbance was set at 5.5 in accordance with previous research [19]. In this study, participants with a PSQI-J score of ≥5.5 were allocated to the “low sleep quality group” and those with <5.5 to the “high sleep quality group”.

#### 2.2.2. Kihon Checklist (KCL)

Developed by the Japanese Ministry of Health, Labor, and Welfare, the KCL serves as a screening tool to identify individuals eligible for secondary prevention interventions at risk of requiring long-term care [12]. The checklist comprises 25 questions over seven domains: (1) activities of daily living, (2) mobility, (3) nutritional status, (4) oral function, (5) confinement, (6) cognitive function, and (7) depressive mood. Each item is scored from 0 to 1 points, leading to a final score range of 0–25 points. In this study, two KCL scores are used: (1) the total KCL score and (2) the KCL score excluding the five depressive mood items. After excluding these items, a score of 10 or more from the remaining 20 items indicates risk of impaired independence. Ogawa and colleagues, [23] in their examination of frailty determination, based on the frailty phenotype developed by Fried et al. [24], found a cutoff value of 5 or more points on the KCL (excluding the 5 depression items) appropriate for indicating frailty.

#### 2.2.3. Conversation

To assess the association between relationships (human interactions, conversations) and sleep among older adults, our survey employed “frequency of conversation” and “discussion partner for important matters” as conversation-related metrics. While previous reports assessed weekly conversation frequency among older, in the current study, conversation was defined as “the number of conversations per day”, considering social interaction’s role in the circadian rhythm entrainment factors [25,26]. Conversation frequency was scored as follows: almost never = 0, once = 1, twice = 2, three times = 3, four times = 4, and five or more times = 5. Additionally, “discussion partner for important matters” was categorized according to whether the person was a spouse or not, considering the potential association between sleep quality and conversation partners, as indicated by Matsuda et al. and others [27,28].

#### 2.2.4. Measurement of Other Variables

The DRACE, comprising 12 items, was designed to evaluate mastication and swallowing function in older adults. Responses to each item are scored on a 3-point scale: not at all = 0, sometimes = 1, and often = 2. A score of 3–5 or higher signifies a risk of dysphagia [26].

The GDS-15-J is the short-form version of the GDS-30 scale used to assess depression in older adults. It features 15 items, with responses of “no” or “yes” scored 0 or 1, respectively. Total scores thus range from 0 to 15. A cutoff of 5–6 points is used as a depression screening scale [29].

The UCLA-J, developed as a loneliness scale for older adults, encompasses 20 questions with responses “always”, “sometimes”, “rarely”, and “never”, scored from 1 to 4, respectively. The total score ranges from 20 to 80 points, with higher scores indicating greater loneliness [30].

#### 2.2.5. Statistical Analysis

For continuous variables, uncorrelated *t*-tests and Mann–Whitney U tests were applied, while the chi squared test was used for categorical variables in univariate analyses. Spearman rank correlation coefficients were calculated for variables displaying statistically significant differences (significance level * *p* < 0.05, ** *p* < 0.01) and for independent variables.

The dependent variable was a binary PSQI-J variable (≥5.5 = 1, <5.5 = 0), and explanatory variables were age, sex, older households, number of diseases, DRACE, GDS-15, UCLA-J, conversation frequency, and discussion partner for important matters. A binomial logistic regression analysis (forced entry method) was conducted. Because the KCL shares some items in common with the DRACE, GDS, and items pertaining to frequency of going out, these variables were excluded to avoid content overlap. A diagnosis of multicollinearity among each independent variable required a variance inflation factor (VIF, VIF = I/(I − R2)), 10 or more for three or more variables.

The Hosmer–Lemeshow test was used to evaluate goodness of fit for the model in the logistic regression analysis. SPSS version 27 (IBM Corporation, Armonk, NY, USA) was used for statistical analysis, and the significance level was set at less than 5%.

## 3. Results

### 3.1. Basic Demographics of Study Participants

As depicted in Table 1, the participants in this study (500 individuals: 250 men and 250 women) were independently living older adults aged 65 years or above. The mean age was 73.57 (SD 5.54) years, approximating the 2019 estimated healthy life expectancy of 74 years. Employed individuals numbered 141 (28.2%), closely resembling the 25.1% employment rate among persons aged 65 and above in Japan [31]. Regarding household composition, 349 (69.8%) comprised older households with an older couple or single individual living alone, in alignment with figures from a national survey [32]. The overall average BMI was 22.6 (SD 3.29) kg/m^2^, which is within the target range for the those aged 65 years and over (21.5–24.9) [33].

### 3.2. Results Overview

Sleep quality was divided into two groups based on PSQI-J scores, with 167 (33.4%) people in the low sleep quality group (i.e., those with poor sleep quality) and 333 (66.6%) in the high sleep quality group (i.e., those with good sleep quality). Significant differences between the two groups were found in scores for DRACE, GDS-15-J, UCLA-J, and KCL (excluding depression-related items), and frequency of outings, conversations, and discussion partner for important matters. Furthermore, the low sleep quality group had a mean KCL (excluding depression items) score of 5.55 (SD 2.47), which corresponds to frailty.

Table 2 shows the correlation coefficients between each factor, which confirmed that the VIF values obtained were not multicollinear. Binary logistic regression analysis with the dependent variable being the binary PSQI-J (≥5.5 = 1, <5.5 = 0) suggested that the factors associated with decreased sleep quality were slightly younger age, frequency of outings, risk of dysphagia, and depression; furthermore, a decrease in the frequency of conversation was found (Table 2). Table 3 shows the respective odds ratios (95% CI (lower limit, upper limit)) were age 0.944 (0.908–0.983), number of diseases 1.469 (1.256–1.719), DRACE 1.105 (1.094–1.182), GDS-15-J 1.156 (1.069–1.25), frequency of conversations 0.879 (0.765–0.895), and discussion partner for important matters (spouse only) 1.821 (1.113–2.978) (Table 3).

The model also showed strong goodness of fit (Hosmer–Lemeshow test *p* = 0.891). Thus, we can conclude the pertinent factors associated with the low sleep quality group in this study were successfully extracted from the physical, mental, and the social aspects of the older adult participants.

## 4. Discussion

Decreased sleep quality was found to be associated with age, number of diseases under treatment, DRACE, GDS-15-J, frequency of conversation, and discussion partner for important matters (spouse only). Given that 30% of older adults in Japan experience insomnia, sleep assessment may contribute to the early detection of frailty associated with the low sleep quality group, these factors should be considered as pertinent from the viewpoint of assessing frailty.

### 4.1. Age-Related Findings

The mean age of the low sleep quality group was 73 (SD 5.73), not significantly different from that of the high sleep quality group, both around 75 years. This age marks a point where medical and long-term care needs rise, aligning with a healthy life expectancy of 74. This pivotal age underscores the importance of assessing older adults’ physical, mental, and social status. Such insights could be helpful for extending healthy life expectancy and enhancing care needs predictability.

Although sleep quality may be affected by lifestyle-related diseases such as hypertension and diabetes [34], this study found no significant differences regarding the type of illness. However, the low sleep quality group had significantly more diseases under treatment. A nationwide Japanese survey (Ministry of Health, Labor, and Welfare, 2017) revealed higher outpatient rates for those aged 70–74 (10.6%) compared to those aged 65–69 (7.8%) [35]. The current study suggests a link between sleep quality and the increased medical visits among older adults, particularly those aged 70 years and above.

### 4.2. Dysphagia Risk

Although the effects of dysphagia during sleep have been noted, few reports have examined sleep quality using DRACE. Maki et al. linked DRACE scores (OR: 1.073) to sleep disturbance through a survey of 400 Japanese older adults (200 requiring nursing care and 200 living independently) [36]. The mean DRACE score was 3.7 (SD 3.8), rising to 4.8 (SD 4.3) in the sleep disorder group (*n* = 174) [36]. Though the current study excluded nursing care-dependent older adults, a significant DRACE score distinction emerged between the two sleep quality groups. The low sleep quality group scored 3.2 (SD 3.75), indicating aspiration risk. Despite our results being slightly lower than those reported by Maki et al., a clear association between aspiration risk and sleep quality was evident.

Aspiration can lead to aspiration pneumonia, which is a major fatality cause among older adults, accounting for more than 70% of pneumonia deaths in those aged 70 years or over. It is estimated that aspiration pneumonia deaths will increase from 38,462 (in 2018) to 120,000 by 2030 [37]. Given this projection, Okazaki et al. highlighted the necessity for new management strategies for dealing with this pressing health concern [38]. The KCL, an oral function item, is a useful item for identifying frailty. The percentage of frail patients successfully identified in a previous survey that used this item was over 40% [39].

Although Fried et al.’s frailty phenotype did not include items on oral function [21], a growing number of studies point to the necessity of aspiration risk assessment [40,41]. Therefore, integrating DRACE to measure aspiration risk alongside holistic care needs assessment instruments like the KCL could provide a more comprehensive depiction of older adults’ care needs and health risks.

### 4.3. Depression

The GDS-15-J score for the low sleep quality group was 4.68 (SD 3.35), slightly below the cutoff value of 5, which is indicative of a depressive tendency. Confirming previous research, our results confirm the association between insomnia and depression. For older adults, this connection holds even more weight due to the serious risk of suicide [42]. In Japan, suicide rates among older adults, especially those aged 80 and above, are high, with a 19%mortality rate compared to the overall 16% [43]. Among those aged 70 and over, health issues and family problems are the primary suicide motives [43]. A conceptual analysis of suicide among Japanese older adults revealed some common antecedents leading to suicide including “illness distress and psychosomatic symptoms”, “vulnerabilities in caring relationships and environment”, “economic problems”, “exceeding the limit of endurance”, and the desire to have “freedom from distress” [44]. “Loss” is also tied to this concept, and it is suggested that suicide “should be considered a moral concern” [44]. While aging brings physical and mental decline, which can contribute to suicide, the decision often involves family dynamics, relationships, and support systems (or lack thereof). In Japan’s aging society, understanding the links between insomnia, depression, and suicide underscores the importance of bolstering the social support environment around older adults.

### 4.4. Conversation

To explore the link between insomnia and social interactions among older adults, our study focused on conversation, a key aspect of human relationships. Our findings revealed that the low sleep quality group engaged in fewer conversations, with respondents considering their spouse as their main “discussion partner for important matters”.

Few studies have probed conversation’s impact on sleep quality. A review of intervention studies (search period 2000–2021) identified two relevant articles, one a randomized control trial, which suggested that conversation, including reminiscence, may enhance sleep quality [45]. Importantly, the need for physiological validation has been highlighted; enjoyable conversation boosts parasympathetic function and promotes sleep [46]. Conversely, some reports link poorer sleep quality to spouse-dominated conversation [27], where the scope of social interaction is often limited. Thus, assessing connections to loneliness and isolation for community-dwelling older couples is crucial.

While sleep-inducing drugs are common for treating insomnia, future research should explore conversational care’s nonpharmacological efficacy, especially in light of side effects of sleep-inducing drugs [47] and the dangers of polypharmacy in older adults [48]. Developing effective conversational care protocols and performing physiological validation are essential steps toward this goal.

### 4.5. Limitations of the Study

It is important to acknowledge the limitations of the current study. Firstly, we adopted a cross-sectional design, which precluded the consideration of the evolving psychosomatic and social effects of the COVID-19 pandemic over time. Second, the study participants were drawn from research monitors who utilized a specific internet-based survey tool. Therefore, exercising caution is essential before attempting to extrapolate the findings to a wider context. Additionally, in the current study, the correlation between conversation and sleep quality, the risk of aspiration, and so on, is solely based on self-reported measures. Consequently, further investigation involving more objective methods is necessary before arriving at any definitive conclusions. Finally, the implications of sleep disorders on the findings of our study were not thoroughly examined, particularly obstructive sleep apnea, which is a significant risk for older adult men [49].

## 5. Conclusions

In our examination of older adults living independently within the community, the outcomes indicate that the individuals allocated to the low sleep quality sleep quality group were already displaying indications of frailty. This signals the potential need for proactive measures to avert progression towards long-term care requirements. Aligned with the Japanese governmental objective of extending healthy life expectancy by a minimum of 3 years by 2040 [50], our research underscores the significance of early detection of frailty among individuals aged 70 years and above. In this context, our study’s findings suggest that evaluating sleep quality alongside its associated factors—such as age, number of diseases under treatment, aspiration risk, depression, and conversation habits—could serve as valuable elements in the initial screening for frailty. Moreover, our observations accentuate the pivotal roles of conversation, companionship, and social interaction in the overall health and wellbeing of older adults.

## Figures and Tables

**Table 1 geriatrics-08-00091-t001:** Comparison of the low and high sleep quality group characteristics in Japanese older adults.

	Entire Cohort	Low Sleep Quality Group	High Sleep Quality Group	*p*-Value
	*n* = 500	(PSQI-J ≥ 5.5) *n* = 167 (33.4%)	(PSQI-J < 5.5) *n* = 333 (66.6%)	
Age (years)	73.57 ± 5.54	73 ± 5.73	73.86 ± 5.43	0.171
Occupation:				
Yes	141 (28.2%)	51	90	0.41
None	359 (71.8%)	116	243	
Household: Older couple Other	258 (51.6%) 242 (48.4%)	76 91	182 151	0.054
BMI	22.6 ± 3.29	22.28 ± 3.09	22.76 ± 3.38	0.101
Outing frequency	3.47 ± 2.36	3.23 ± 2.35	3.59 ± 2.36	0.118
Number of diseases under treatment	1.16 ± 1.32	1.61 ± 1.49	0.93 ± 1.16	<0.001 **
DRACE	2.34 ± 3.07	3.2 ± 3.75	1.92 ± 2.57	<0.001 **
GDS-15-J	3.49 ± 3.11	4.68 ± 3.35	2.89 ± 2.8	<0.001 **
UCLA-J	40.25 ± 10.26	42.05 ± 10.21	38.71 ± 10.07	<0.001 **
Frequency of conversation/day	3.78 ± 1.67	1.78 ± 1.56	4.22 ± 1.33	<0.001 **
Discussion partner for important matters: Spouse only Other	192 (38.4%) 308 (61.6%)	50 117	142 191	0.006 *
KCL	5.71 ± 3.29	6.89 ± 3.55	5.11 ± 2.98	<0.001 **
KCL (excluding depression items)	4.88 ± 2.455	5.55 ± 2.47	4.55 ± 2.38	<0.001 **

Binomial values are listed as *n* (%) or mean ± standard deviation; *p*-value: low sleep quality group vs. high sleep quality group; Mann–Whitney U test, chi squared test, * *p* < 0.01, ** *p* < 0.001; PSQI-J: Pittsburgh Sleep Quality Index—Japan; DRACE: Dysphagia Risk Assessment for Community-Dwelling Elderly; GDS-15-J: The Geriatric Depression Scale 15—Japan; UCLA-J: University of California, Los Angeles Loneliness Scale—Japan; KCL: Kihon Check List.

**Table 2 geriatrics-08-00091-t002:** Correlation of independent variables for sleep among older adults.

.	Age	No. of Diseases	DRACE	GDS-15-J	UCLA-J	Older Households	Frequency of Conversation/Day	Discussion Partner for Important Matters	KCL (Excluding Depression Items)	PSQI-J
Age	―									
Number of diseases	0.123 **	―								
DRACE	0.086	0.122 **	―							
GDS-15-J	−0.054	0.117 **	0.280 **	―						
UCLA-J	−0.126 **	0.005	0.170 **	0.492 **	―					
Elderly households	0.045	0.039	0.070	−0.160 **	−0.143 **	―				
Frequency of conversation/day	0.042	−0.084	−0.001	−0.149 **	−0.275 **	0.334 **	―			
Discussion partner for important matters	−0.099 *	−0.011	0.006	−0.152 **	−0.125 **	0.583 **	0.455 **	―		
KCL (excluding depression items)	0.034	0.234 **	0.429 **	0.324 **	0.237 **	−0.089 *	−0.110 *	−0.087	―	
PSQI	−0.111 *	0.293 **	0.294 **	0.304 **	0.158 **	−0.100 *	−0.215 **	−0.172 **	0.269 **	―

Spearman’s rank correlation coefficient * *p* < 0.05, ** *p* < 0.01. The VIF values calculated from the correlation coefficients between each factor confirmed that there was no multicollinearity among variables (VIF values < 10).

**Table 3 geriatrics-08-00091-t003:** Factors associated with poor sleep quality (low sleep quality group) among independently living Japanese older adults.

	Odds Ratio	95%CI		*p*-Value
		Lower Limit (LL)	Upper Limit (UL)	
Age	0.944	0.908	0.983	0.005 **
Elderly couples	0.88	0.541	1.43	0.606
Number of Diseases	1.469	1.256	1.719	<0.001 ***
DRACE	1.105	1.094	1.182	0.003 **
GDS-15-J	1.156	1.069	1.25	<0.001 ***
UCLA-J	0.897	0.973	1.022	0.822
Frequency of conversation/day	0.879	0.765	0.895	0.042 *
Discussion partner for important matters: Spouse only	1.821	1.113	2.978	0.017 **

Binomial logistic regression analysis model chi squared test * *p* < 0.05, ** *p* < 0.01, *** *p* < 0.001; Hosmer–Lemeshow’s test and chi squared test = 12.347, *p* = 0.136. Judgmental success rate = 71%; compulsory input method; Dependent Variable: low sleep quality group (PSQI-J ≥ 5.5): 1; high sleep quality group (PSQI-J < 5.5): 2.

## Data Availability

The data presented in this study are available in Table 1, Table 2 and Table 3.

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
