# Peer review of "Frailty and Diminished Human Relationships Are Associated with Poor Sleep Quality in Japanese Older Adults: A Cross-Sectional Study"

_geriatrics, 2023, doi:10.3390/geriatrics8050091_

Round 1
Reviewer 1 Report
General
The manuscript by Ben Amor and colleagues aimed to evaluate the association between sleep quality, frailty, and human relationships in a sample of older adults in Japan.
The introduction is long and confusing. I suggest that the authors can start by describing social changes, then social fragility, and a description of the instruments of interest. This change would give a better structure before the study’s aim. Further, the authors should rephrase the purpose of the study.
The lack of justification for the sample size is striking in Methods. What was the variable used to calculate the sample size? Please detail the calculation.
It is necessary to indicate on which dates the investigation was carried out.
The last two paragraphs of the Results section should be incorporated into the Discussion, where appropriate.
The Discussion is also extensive and covers many factors possibly related to the topic. I suggest shortening this section. In addition, this should begin by briefly summarizing the main findings found.
The conclusion should be toned down since the authors only studied one association (e.g., they did not allocate the groups as indicated).
Minors:
- In the Abstract: define KCL
- In Methods:
What is the UCLA-J? (line 107)
Include citations (lines 115 and 119)
Authors should refer to both groups as “low-sleep quality” and “high-sleep quality”
- In Results:
lines 181-184 are redundant
line 201 and elsewhere: please indicate “associated with…” or similar, as this study is merely descriptive
line 203: define LL and UL in the text.
Overall, the manuscript is well written, but the English language must be improved.
Author Response
Reviewer 1
Thank you very much for your detailed and very helpful review of our manuscript. We have responded to all of your comments and have made every effort to improve it based upon your invaluable comments. Below follows a point-by-point response to each comment displayed as a table.
|
Comment |
Response |
Manuscript |
|
1. The introduction is long and confusing. I suggest that the authors can start by describing social changes, then social fragility, and a description of the instruments of interest. This change would give a better structure before the study’s aim. |
Thank you for this comment and we apologize that the introduction came across as confusing. We have rewritten and rearranged the introduction according to the reviewer’s suggestions. We hope it is now more logical in flow. |
Extensive changes to the introduction. |
|
2. Further, the authors should rephrase the purpose of the study. |
We have rephrased the purpose of the study as suggested by the reviewer. |
“Given the need to accurately assess the care needs of older adults and the intricate web of factors that can influence their health well-being, the objective of the current study was to investigate the interplay between conversation and other elements that are associated with sleep quality. This study targeted independently-living Japanese older adults with the intention of contributing to the prevention of long-term care dependency. Recognizing the evolving social dynamics and the multifaceted nature of frailty, the study sought to investigate potential connections between conversation habits and sleep quality among this demographic.”
|
|
3. The lack of justification for the sample size is striking in Methods. What was the variable used to calculate the sample size? Please detail the calculation.
|
Thank you for this comment. We have added a detailed description about how we came to choose 500 as the sample size for our study. |
The sample size was determined by first considering the elderly population using the Internet. In Fiscal Year 2021, Internet usage among individuals aged 65 and above was approximately 53.4% [18]. Utilizing this percentage, along with a 95% confidence level and a 5% margin of error, the required population proportion was calculated to be 382 participants. Furthermore, the statistical analyses for this study involved t-tests with men and women aged 65 and older. Consequently, by assuming a moderate effect size (d = 0.5), an alpha of 0.05, a power of 0.95, and an allocation ratio of 1, each group necessitated 105 participants (210 in total) (using G Power). Consequently, a sample size of 500 was established, surpassing all criteria. |
|
3. It is necessary to indicate on which dates the investigation was carried out.
|
Thank you for this important comment. We have added the relevant information as you suggested. |
“The survey was carried out from June 1 to 20, 2021.”
|
|
4. The last two paragraphs of the Results section should be incorporated into the Discussion, where appropriate.
|
Thank you for this suggestion. We have incorporated the last two paragraphs into the discussion, or removed them where appropriate. |
Changes made to Results and Discussion sections. |
|
5. The Discussion is also extensive and covers many factors possibly related to the topic. I suggest shortening this section.
|
Thank you for this suggestion. We reduced the length of this section by approximately 20%. |
Extensive changes to the discussion. |
|
6. In addition, this should begin by briefly summarizing the main findings found. |
We added a brief summary to the discussion as you suggest. |
Summary added |
|
7. The conclusion should be toned down since the authors only studied one association (e.g., they did not allocate the groups as indicated).
|
We tried to tone down the conclusion and also rewrote and expanded the preceding limitation section. |
Multiple changes to the conclusion. |
|
8. - In the Abstract: define KCL
|
Thank you for this comment. We added a description of the KCL as you suggested. |
“…Kihon Checklist (a Japanese instrument used to determine the care needs and frailty of older adults),…” |
|
9. – In Methods: What is the UCLA-J? (line 107) |
Thank you for this comment. We added a description of the UCLA-J as you suggested. |
“…and the Japanese version of University of California Los Angeles Scale (UCLA-J), which is an instrument to determineloneliness in older adults.” |
|
10. Include citations (lines 115 and 119)
|
Citations have been added to the indicated sections. |
Citations [19] and [12]. |
|
11. Authors should refer to both groups as “low-sleep quality” and “high-sleep quality”
|
Thank you for this comment. We have revised the group names as you have suggested. |
Multiple changes throughout the manuscript. |
|
12. - In Results: lines 181-184 are redundant
|
We agree that these lines are redundant and have removed them. |
Text indicated as redundant by the reviewer has been removed. |
|
13. line 201 and elsewhere: please indicate “associated with…” or similar, as this study is merely descriptive
|
We have attempted to revise the manuscript at various points in accordance with your important suggestion. |
Manuscript changes at various points. |
|
14. line 203: define LL and UL in the text.
|
Thank you for pointing this out, we have given the definitions here as you suggested. |
“Table 3 shows the respective odds ratios (95% CI [lower limit, upper limit]) were: …” |
|
15. Overall, the manuscript is well written, but the English language must be improved.
|
Thank you. We have attempted to improve the English language throughout the manuscript. |
Extensive changes to the English language of the manuscript. |
Reviewer 2 Report
The authors present interesting, well-analyzed, novel findings linking age-related frailty to sleep quality in Japanese older adults. The paper provides sufficient introduction into the issue of frailty and current knowledge gaps. The findings speak for themselves. I have only minor suggestions to improve the readability of the manuscript, as listed below:
Abstract
-The abstract is missing the definition of the acronym KCL. I assume this is the Kihon Checklist, but it is not defined.
Introduction:
-Lines 39-40: multiple use of the word "however" confuses the purpose of the sentence.
-Lines 81-84: The stated aim of the current study does not accurately capture the scope of the analyses. I suggest rephrasing the aim in broader terms.
Discussion:
-Lines 253-254: "Although the effects of aspiration of saliva during sleep have been noted, there have 253 been few reports examining sleep quality using DRACE". The mention of saliva aspiration during sleep is out of context. It took several reads to understand that the authors are trying to say that individuals with dysphagia may choke on their own saliva. I suggest rephrasing to avoid confusion. This discussion point is also out of context since the results do not report any findings directly related to drooling/aspiration. If the DRACE has a specific item related to drooling that was related to sleep quality, this sub-analysis should be included in the Results if it merits discussion.
Limitations: the limitations section is anemic. Offhand, multiple other limitations can be found just by reading the paper: the tenuous link between conversation and sleep quality; the lack of data with regards to aspiration risk, the lack of objective sleep or health data and the Japanese fluency language requirement to name a few.
-I recommend referring to the sleep quality groups as "poor sleep quality" and "good sleep quality" rather than "low" and "high". Higher PSQI scores relate to poorer sleep quality, so using directional terms like "low" and "high" can be confusing to the reader.
There are some terms or use of English phrases throughout that are confusing upon first reading. These are mistakes that a native speaker could make and does not reflect the language skills of the writer. However, it is sometimes difficult for the original writer to identify when a sentence sounds awkward or fails to convey meaning. I suggest having another author or member of the research team give the manuscript a once-over to ensure that they can follow the story.
Author Response
Reviewer 2
Thank you very much for your detailed and very helpful review of our manuscript. We have responded to all of your comments and have made every effort to improve it based upon your invaluable comments. Below follows a point-by-point response to each comment.
|
Comment |
Response |
Manuscript |
|
1. -The abstract is missing the definition of the acronym KCL. I assume this is the Kihon Checklist, but it is not defined.
|
Thank you for this comment and we apologize for the oversight. We have added the acronym and also a brief description of the Kihon Checklist. |
“…Kihon Checklist (KCL), a Japanese instrument used to determine the care needs and frailty of older adults,…” |
|
2. -Lines 39-40: multiple use of the word "however" confuses the purpose of the sentence.
|
Thank you for pointing this out. It was indeed very confusing. Based on the suggestion of another reviewer we have restructured and rewritten the introduction, and revised the issue that you pointed out has been resolved. |
“The Kihon Checklist (KCL), developed by the Japanese Ministry of Health, Labour and Welfare [12] serves as a reliable screening tool to identify high-risk individuals necessitating nursing care[13,14]. While the KCL's effectiveness in assessing physical and mental frailty in older adults is widely acknowledged, [15] it lacksan assessment of sleep quality.” |
|
3. -Lines 81-84: The stated aim of the current study does not accurately capture the scope of the analyses. I suggest rephrasing the aim in broader terms.
|
Thank you for this comment. We agree with you and have rephrased the aim of the study. We hope that it is now clearer. |
“Given the need to accurately assess the care needs of older adults and the intricate web of factors that can influence their health well-being, the objective of the current study was to investigate the interplay between conversation and other elements that are associated with sleep quality. This investigation targeted independently-living Japanese older adults with the intention of contributing to the prevention of long-term care. dependency. Recognizing the evolving social dynamics and the multifaceted nature of frailty, the study sought to shed light on the potential connections between conversation habits and sleep quality among this demographic.”
|
|
4. -Lines 253-254: "Although the effects of aspiration of saliva during sleep have been noted, there have 253 been few reports examining sleep quality using DRACE". The mention of saliva aspiration during sleep is out of context. It took several reads to understand that the authors are trying to say that individuals with dysphagia may choke on their own saliva. I suggest rephrasing to avoid confusion. This discussion point is also out of context since the results do not report any findings directly related to drooling/aspiration. If the DRACE has a specific item related to drooling that was related to sleep quality, this sub-analysis should be included in the Results if it merits discussion. |
Thank you for this important comment. We have eliminated the mention of saliva aspiration as the reviewer suggests and have mentioned only dysphagia in general. We hope that this resolves the reviewer’s concerns. |
“Although the effects of dysphagia during sleep have been noted, few reports have examined sleep quality using DRACE.” |
|
5. Limitations: the limitations section is anemic. Offhand, multiple other limitations can be found just by reading the paper: the tenuous link between conversation and sleep quality; the lack of data with regards to aspiration risk, the lack of objective sleep or health data and the Japanese fluency language requirement to name a few.
|
Thank you for this observation. We agree with the reviewer about the limitations section and have attempted to rewrite it in light of the reviewer’s comment. We hope it not better reflects the limitations of the study. |
Limitations section substantially revised.
|
|
6. -I recommend referring to the sleep quality groups as "poor sleep quality" and "good sleep quality" rather than "low" and "high". Higher PSQI scores relate to poorer sleep quality, so using directional terms like "low" and "high" can be confusing to the reader.
|
Thank you for your valuable suggestions. We can certainly see the merit of using “good” and “poor”. Interestingly, another reviewer made the opposite suggestion that we use “low-sleep quality group” and high-sleep quality group.” Thus, we added a description to section 3.2 (Results overview) using the terms poor and good to clarify the meaning, and have used “low-sleep quality group” and high-sleep quality group” consistently throughout the manuscript for clarity. We hope that this satisfies the reviewer’s concerns. |
“Sleep quality was divided into two groups based on PSQI-J scores, with 167 (33.4%) people in the low-sleep quality group (i.e., those with poor sleep quality) and 333 (66.6%) in the high-sleep quality group (i.e., those with good sleep quality).” |
|
There are some terms or use of English phrases throughout that are confusing upon first reading. These are mistakes that a native speaker could make and does not reflect the language skills of the writer. However, it is sometimes difficult for the original writer to identify when a sentence sounds awkward or fails to convey meaning. I suggest having another author or member of the research team give the manuscript a once-over to ensure that they can follow the story.
|
Thank you for this very true observation. The second author, a native English speaker, who worked extensively on the manuscript, was reluctant to alter the voice of the first author too much. However, we decided, based on the reviewer’s suggestion (echoed also by another reviewer) to work further on the English and hope that it is improved to a satisfactory level. |
Extensive English editing of the manuscript. |
Reviewer 3 Report
This is an very useful article for how to deal in a very proactive and clinically meaningful way with the challenges of aging .
I wish that the refs in Japanese [ie Re 38 ] would also be available in English .
Author Response
Reviewer 3:
Thank you for your positive response to our manuscript. We are happy that you feel it worthy of publication. We have responded to your comment about the references below.
|
Comment |
Response |
Manuscript |
|
I wish that the refs in Japanese [ie Re 38 ] would also be available in English .
|
Thank you for this important comment. We have taken some time to reflect upon it and agree that it would indeed be good if these references were available in English. We had to choose the most relevant data to address our population, so it’s somewhat unavoidable that the references are in Japanese. However, in response to the reviewer’s comment, we have added further references to studies published in English that we hope will support the text. In addition, we believe hat advances in AI and translation software will allow those interested in the details of these references to access this information. Moreover, we have attempted to contextualize the relevant research and interpret it within our paper and hope that it makes these Japanese studies more accessible to a wider audience. |
Citations to English-language publications added. |
Reviewer 4 Report
geriatrics-2549513
Frailty and Diminished Human Relationships are Associated with Poor Sleep Quality in Japanese Older Adults: A Cross-sectional Study
This is a very thought provoking paper. There is no question that the investigators are leading us into an important area: The relationship of sleep quality to factors, such as frailty in older adults. The clinical utility and relevance cannot be understated. This said, the paper would benefit from the following:
Strengths
· Very good and important foundational question
· Great rationale for the study and is clinical utility
· Good size sample
Weaknesses
· Introduction: The introduction would benefit from re-organization. In other words, it would helpful if the others introduced concepts, such as frailty, earlier in the introduction. As it stands right now, these concepts are not introduced until the end, which can be a little confusing. It would make more sense to move the discussion re. the KCL to the end and move the other concepts to the beginning. Finally, please make sure that all terms are appropriately operationalized (e.g., insomnia vs sleep quality)
· Methods: Given that Appendix A doesn’t contain a ton of information, it would be better if the authors included all of that information in the main body of the text.
· Intro/Discussion: The authors state, “Although the participants in the current study did not include older adults requiring nursing care, a significant difference in DRACE scores was found between the
two sleep quality groups, with scores of 3.2 (SD 3.75) in the low group being in the range
to indicate risk for aspiration. Although our results were 1 point lower in each group than
that those reported by Maki et al., we were able to identify a clear association between
aspiration risk and sleep quality. Aspiration can lead to aspiration pneumonia, which is a
major leading cause of death, accounting for more than 70% of pneumonia deaths in older
adults aged 70 years or over.” While this may be true, did these subject endorse any symptoms related to OSA on the PSQI? This age group, especially men, are also at risk for OSA. It is not a new concept that having difficulty breathing at night will impact sleep quality. Please include a conversation about the risk for other sleep disorders in the discussion, particularly in limitations if you didn’t assess for them. In addition, an overall conversation in the introduction re. what you mean by “sleep quality” may be warranted as this is a broad term that can mean many things. The authors discuss insomnia but then switch over to talking about sleep quality, which aren’t the same.
· Please include Cronbach’s alphas for all study questionnaires.
· Please include a statement on the IRB approval for this study.
Author Response
Reviewer 4
Thank you very much for your detailed and very helpful review of our manuscript. We have responded to all of your comments and have made every effort to improve it based upon your invaluable comments. Below follows a point-by-point response to each comment.
|
Comment |
Response |
Manuscript |
|
1. The introduction would benefit from re-organization. In other words, it would helpful if the others introduced concepts, such as frailty, earlier in the introduction. As it stands right now, these concepts are not introduced until the end, which can be a little confusing. It would make more sense to move the discussion re. the KCL to the end and move the other concepts to the beginning. Finally, please make sure that all terms are appropriately operationalized (e.g., insomnia vs sleep quality)
|
Thank you for pointing this out. Based on the suggestion of yourself and another reviewer we have restructured and rewritten the introduction extensively. |
Extensive restructuring and rewriting of the introduction. |
|
2. Given that Appendix A doesn’t contain a ton of information, it would be better if the authors included all of that information in the main body of the text.
|
Thank you for this suggestion. As you suggested we have removed Appendix A and included the relevant information in the main body of the text. We have left the second part of the Appendix as the first author believes this will be of interest to some of the readers and important to the study. |
“Participant informed consent was obtained prior to conducting the research. Participants were required to abide by the personal information protection regulations (https://insight.rakuten.co.jp/company/privacy.html), and meet the following inclusion conditions: (1) no “care-needs certification” requiring 1-5 years of nursing care, (2) absence of a dementia diagnosis, and (3) capacity to comprehend and act upon instructions and responses. To protect the participants’ personal information, Rakuten Insightprovided anonymized participant data to the principal investigator and acted as an intermediary when participants wished to withdraw consent or had inquiries about the study.” |
|
3. While this may be true, did these subject endorse any symptoms related to OSA on the PSQI? This age group, especially men, are also at risk for OSA. It is not a new concept that having difficulty breathing at night will impact sleep quality. Please include a conversation about the risk for other sleep disorders in the discussion, particularly in limitations if you didn’t assess for them.
|
Thank you for this comment. Obstructive Sleep Apnea (OSA), as with other sleep disorders, was not addressed specifically in this study. But, as the reviewer has pointed out, this is a really important topic, and have revised the text of the limitations to reflect the reviewer’s comment. |
“Finally, the implications of sleep disorders on the findings of our study were not thoroughly examined, particularly obstructive sleep apnea, which is a significant risk for older adult men [49].” |
|
In addition, an overall conversation in the introduction re. what you mean by “sleep quality” may be warranted as this is a broad term that can mean many things. The authors discuss insomnia but then switch over to talking about sleep quality, which aren’t the same.
|
Thank you for this comment. We have restructured and rewritten parts of the introduction, and hope that the improved flow of ideas, plus more clarity over the definition of the two study groups satisfies the reviewer’s comment here. |
Extensive changes to the introduction. |
|
·Please include Cronbach’s alphas for all study questionnaires.
|
Thank you for this comment. We have added Cronbach’s alphas for the DRACE, GDS-15, PSQI-J, and UCLA-J. |
“The validity of the PSQI-J [19], DRACE [20], GDS-15-J [21], and UCLA-J [22], has been supported in previous validation studies by Cronbach’s alpha coefficients of 0.77, 0.88, and 0.84, respectively.” |
|
·Please include a statement on the IRB approval for this study.
|
The ethical statement that was included in the original manuscript was probably temporarily removed by the editorial team during the peer review process. However we have included it in the main body of the text as the reviewer has suggested. |
The survey was conducted from June 1 to 20, 2021, in accordance with the guidelines of the Declaration of Helsinki and after being approved by the ethics committee of the Institute of Medicine, University of __________ (Approval No. ________).” |